# Extracellular Histones Activate Plasma Membrane Toll-Like Receptor 9 to Trigger Calcium Oscillations in Rat Pancreatic Acinar Tumor Cell AR4-2J

**DOI:** 10.3390/cells8010003

**Published:** 2018-12-20

**Authors:** Hai Yan Guo, Zong Jie Cui

**Affiliations:** Institute of Cell Biology, Beijing Normal University, Beijing 100875, China; guohaiyan111@163.com

**Keywords:** extracellular histones, TLR9, calcium oscillations, AR4-2J

## Abstract

In acute pancreatitis, histones are released by infiltrating neutrophils, but how histones modulate pancreatic acinar cell function has not been investigated. We have examined histone modulation of rat pancreatic acini and pancreatic acinar tumor cell AR4-2J by calcium imaging. Histones were found to have no effect on calcium in pancreatic acini but blocked calcium oscillations induced by cholecystokinin or acetylcholine. Both mixed (Hx) and individual (H1, H2A, H2B, H3, H4) histones induced calcium oscillations in AR4-2J. RT-PCR and Western blot verified the expression of histone-targeted Toll-like receptor (TLR) 2, 4 and 9. Immunocytochemistry identified TLR2/TLR4 on apical plasma membrane and TLR9 in zymogen granule regions in pancreatic acini. TLR2 was found on neighboring and TLR9 on peripheral plasma membranes, but TLR4 was in the nucleus in AR4-2J clusters. Neither TLR2 agonist zymosan-A nor TLR4 agonist lipopolysaccharide had any effect on calcium, but TLR9 agonist ODN1826 induced calcium oscillations; TLR9 antagonist ODN2088 blocked H4-induced calcium oscillations in AR4-2J, which also disappeared after treatment of AR4-2J with glucocorticoid dexamethasone, with concurrent TLR9 migration from plasma membrane to cell interiors. TLR9 down regulation with siRNA suppressed H4-induced calcium oscillations. These data together suggest that extracellular histones activate plasma membrane TLR9 to trigger calcium oscillations in AR4-2J cells.

## 1. Introduction

Neutrophil infiltration is a hallmark of acute pancreatitis [1,2,3,4]; enhanced neutrophil content is also found in pancreatic cancer [5]. Although neutrophil infiltration is a general feature in pancreatic pathology, its possible modulation of pancreatic acinar cell function has not been investigated in detail. We have previously found that neutrophil respiratory burst could completely suppress cholecystokinin 1 (CCK1) or muscarinic 3 (M3) acetylcholine (ACh) receptor-mediated calcium oscillations in the freshly isolated rat pancreatic acini [6]. Another vital function of infiltrated neutrophils is to form extracellular traps, but what effects the individual components of neutrophil extracellular traps (NET) would have on rat pancreatic acinar cells have not been established. Histones are major components of nucleosomes and of NET [7,8]; whether free extracellular histones would have any effect on calcium signaling in rat pancreatic acinar cells is not known.

Histones H2A, H2B, H3, and H4 form the central octameric protein core of nucleosome, with a linker H1 on the hilum [9,10]. Lys-rich H1, H2A, H2B, and Arg-rich H3 and H4 are all cationic proteins at physiological pH, with isoelectric points of H4 (11.3) > H3 (11.02) > H2A (10.95) > H1 (10.11) > H2B (9.76) [11]. The normally nucleus-confined, DNA-bound histones escape into extracellular fluid when phagocytes undergo NETosis to form extracellular traps [12], or otherwise when parenchymal cells die and cell membranes break down [13,14]; from the extracellular space, histones may enter the general circulation.

Circulating histones in the blood in healthy subjects are rather low (0.79–2.30 mgL^−1^) [15,16,17], but could rise steeply up to 230 mgL^−1^ in sepsis [16,18], sterile inflammation [17,19,20,21] or in severe trauma [15]. Circulating histones could also increase to 149.6 mgL^−1^ in mouse necrotizing acute pancreatitis, for example [20]. Increased serum histones (5.9–33.8 mgL^−1^) early after the onset of pancreatitic abdominal pain were correlated to organ failure and mortality; note that circulating histones were barely detectable in healthy volunteers [22]. Further, tail-vein injected histones have been found to home-in to the pancreas in mice [23].

Extracellular histones have been found to stimulate calcium entry in multiple cell types [15,24], to stimulate cytokine synthesis/release [14,25,26,27,28,29] and to inhibit macrophage phagocytosis [30]. Histone injections in mice increased circulating cytokines [21]. Intra-scrotal histone injection has been found to induce trans-endothelial neutrophil migration in skeletal muscles [14]. Extracellular histones therefore possess distinctive activities both in vitro and in vivo.

Extracellular histones have been found to target Toll-like receptors 2, 4 and 9 (TLR2, 4 and 9) in different cell types [14,19,21,25,29]. Histones by activating TLR2 and 4, for example, induce glomerular cell injury [31], and by activating TLR9 induce hepatic reperfusion injury [19]. TLR2 and TLR4 are selectively activated by yeast cell wall component zymosan A [32,33,34] and bacterial cell wall component lipopolysaccharide (LPS) [35,36] respectively. TLR9, in most cases located intracellularly, is activated by viral and bacterial un-methylated CpG-DNA [37,38], as well as by endogenous mitochondrial DNA from mammalian cells [39].

Based on the above facts, we have in the present work investigated the effects of extracellular histones on calcium signaling in the freshly isolated rat pancreatic acini and in cultured rat pancreatic acinar tumor cell AR4-2J. Extracellular histones were found to have no effect on basal (i.e., resting) calcium but blocked CCK- and ACh-induced calcium oscillations in isolated rat pancreatic acini. Both mixed and individual histones were found to activate peripheral plasma membrane TLR9 to trigger calcium oscillations in AR4-2J cell clusters. To the best of our knowledge histone activation of plasma membrane TLR9 to induce calcium oscillations has not been reported before in any cell type, especially in rat pancreatic acinar cells. Immediate histone activation of calcium signaling is different from the longer term effects of altered gene expressions mediated by the canonical MyD88 or Myddosome pathways [19,40,41]. This work also further highlights the importance of TLR9 shuttling between the plasma membrane and the cell interiors which occur due to epithelial cell transformation.

## 2. Materials and Methods

### 2.1. Materials

Sulfated cholecystokinin octapeptide (CCK) was from Tocris Cookson (Bristol, UK). Acetylcholine chloride (ACh), dexamethasone (Dex), zymosan A (TLR2 agonist) and lipopolysaccharides (LPS) from *Escherichia coli* O55:B5 (L2637, TLR4 agonist) were purchased from Sigma-Aldrich (St Louis, MO, USA). Cell-Tak was from BD Biosciences (Bedford, MA, USA). Fura-2 AM was from AAT Bioquest (Sunnyvale, CA, USA). Recombinant histones H1, H2A, H2B, H3, H4 were from New England Biolabs (Boston, MA, USA). Goods buffer 4-(2-hydroxyethyl)-1-piperazineethane-sulphonicacid (HEPES) was from Boehringer Mannheim (Mannheim, Germany). MEM amino acid mixture (50×), DMEM/F12, 0.25% trypsin/EDTA were from Gibco Life Technology (Shanghai, China). TLR9 agonist OND1826 and TLR9 antagonist ODN2088 were from InvivoGen (San Diego, CA, USA). Hoechst 33342 was from DojinDo (Beijing, China). Collagenase P, mixed histones (Hx, cat. no. 10223565001) of calf thymus were from Roche (Mannheim, Germany). Rabbit anti-TLR2 polyclonal antibody (TLR2, H-175, sc-10739) and rabbit anti-TLR4 polyclonal antibody (TLR4, H-80, sc-10741) were from Santa Cruz Biotechnology (Santa Cruz, CA, USA). Mouse anti-TLR9 monoclonal antibody (ab134368) and secondary antibodies (donkey anti-rabbit IgG against TLR2,4 primary antibodies-ab6799, goat-anti-mouse IgG against TLR9 primary antibody-ab6786, all TRITC-labeled) were from Abcam (Cambridge, UK). Top 10 competent cells were from TianGen Biochemicals (Beijing, China). PrimeStar GXL DNA polymerase was from Takara Clontech (Beijing, China).

### 2.2. Isolation of Rat Pancreatic Acini and Culture of AR4-2J Cells

Rat pancreatic acini were isolated as reported previously [6,42,43,44]. Briefly, rat of the Sprague - Dawley strain (250–450 g) was killed by CO_2_ asphyxia. The pancreas was excised and digested with collagenase P (0.2 gL^−1^). The pancreatic acini isolated were washed three times and re-suspended before use. This procedure was approved by the Animal Ethics Committee (CLS-EAW-2017-015) at Beijing Normal University School for Life Sciences.

Buffer for acini isolation had the following composition (in mM): NaCl 118, KCl 4.7, CaCl_2_ 2.5, MgCl_2_ 1.13, NaH_2_PO_4_ 1.0, D-glucose 5.5, HEPES 10, L-glutamine 2.0, and BSA 2%, MEM amino acid mixture 2%, soybean trypsin inhibitor 0.1 gL^−1^. Buffer pH was adjusted to 7.4 with NaOH 4 M.

AR4-2J cell line was purchased from American Type Culture Collection (Rockville, MD, USA) and cultured in DMEM/F12 supplemented with 20% fetal bovine serum and antibiotics in a CO_2_ incubator with 5% CO_2_/95% air as reported before [6,45,46,47].

### 2.3. Reverse Transcription-PCR (RT-PCR)

Total RNA was prepared using TRIzol reagent (Invitrogen) and was reverse transcribed, the resulting cDNA was subject to polymerase chain reaction (PCR). Forward and reverse primers for TLR2, TLR4, and TLR9 were 5′-CGCTTCCTGAACTTGTCC-3′, 5′-GGTTGTCACCTGCTTCCA-3′; 5′-GCCGGAAAGTTATTGTGGTGGT-3′, 5′-ATGGGTTTTAGGCGCAGAGTTT-3′; 5′-GCTTGATGTGGGTGGGAATT-3′, 5′-CCGCCTCGTCTGCCTTTT-3′ respectively. GAPDH (GAPDH primers 5′-GTGGAGTCTACTGGCGTCTT-3′, 5′-CCAGGATGCCCTTTAGTG-3′) was used as an internal control. PCR proceeded with primer pairs for GAPDH, TLR2, TLR4 or TLR9, before agarose gel electrophoresis and imaging.

### 2.4. TLR9 siRNA Knock Down

AR4-2J cultured in DMEM/F12 plus 20% FBS at a confluence of 65–75% were transfected with siRNA. The siRNA transfection agent X-tremeGENE siRNA (10 μL) was first diluted in 90 μL Opti-MEM, 10 μL siRNA-*TLR9* diluted in 90 μL Opti-MEM, before the diluted solutions were mixed. The mixture was added to a 6-well plate with each well containing 1.8 mL DMEM/F12; the medium was replaced with fresh medium 6–8 h later. Transfected cells were used 24 h after transfection. Negative controls were transfected with scrambled sequence (*NC*). siRNA used were as follows: siRNA-*TLR9*-(5′–3′) CCACUUGUCCCUUAAGUAUTT, AUACUUAAGGGACAAGUGGTT; NC-UUCUCCGAACGUGUCACGUTT, ACGUGACACGUUCGGAGAATT.

### 2.5. Western Blot

Isolated rat pancreatic acini or AR4-2J were lysed in lysis buffer (TRIS-HCl 50 mM, NaCl 150 mM, NP-40 1%, SDS 0.1%, pH 7.4; or mammalian protein extraction reagent from Kanweishiji, Beijing, China) containing phenylmethanesulfonyl fluoride (PMSF, 1 mM). Lysates were centrifuged (4 °C, 12,000 *g*, 10 min), and lysate protein concentration was determined. Proteins were separated by SDS-PAGE, blotted onto PVDF and incubated with primary rabbit anti-TLR2, rabbit anti-TLR4 or mouse anti-TLR9. The PVDF membrane was then incubated with HRP-conjugated secondary antibody (Ab) at room temperature for 60 min, imaged and quantified by enhanced chemiluminescence as reported before [44].

### 2.6. Immunocytochemistry

Pancreatic acini attached to Cell-Tak-coated cover-slips or AR4-2J cells grown on cover-slips for 2 days were washed in phosphate-buffered saline, fixed in paraformaldehyde 4% and permeabilized in Triton X 0.2%; non-specific binding was blocked in 3% BSA before incubation with the primary antibody in a humid chamber at 4 °C overnight and then with secondary Ab (40 min) at room temperature. The nuclei were counter-stained with Hoechst 33342. The cover-slips were sealed and stored at 4 °C before fluorescence imaging in a confocal microscope (Zeiss LSM 700). Confocal images were taken with λ_ex_ 555 nm (TRITC) or 405 nm (Hoechst 33342), λ_em_ 576 nm, 461 nm respectively.

### 2.7. Measurement of Cytosolic Calcium Concentration

Freshly isolated rat pancreatic acini were loaded with Fura-2 AM (10 μM, 30 min) in a shaking waterbath, attached to Cell-Tak-coated glass bottom of Sykes-Moore chambers, before perfusion at 1 mL·min^−1^. AR4-2J cells were cultured on glass cover-slips for 2 days; the cell-attached cover-slips were assembled in Sykes-Moore chambers before AR4-2J cells were loaded with Fura-2 AM (10 μM, 60 min) and perifused at 1 mL·min^−1^. Fluorescent calcium imaging was done as reported previously [6,44,47,48] in a Photon Technology International (PTI, Ediosn, N.J., USA) calcium measurement system, with a photon-counting PMT (PTI PMT814) or a CCD (NEO-5.5-CL-3, Andor) as detector, in an inverted fluorescent microscope (Olympus IX or Nikon NE 300), with fluorescence collected at 510 ± 25 nm. The fluorescent ratios of F_340_/F_380_ were shown as indications of calcium concentrations and were plotted against time with SigmaPlot as before [6,47,49].

### 2.8. Data Presentation and Analysis

All data were presented as mean ± SEM; statistical significance was analyzed by Student’s T test, with *P* < 0.05 taken as statistically significant as indicated by an asterisk (*).

## 3. Results

### 3.1. Extracellular Histones Block CCK- and ACh-Induced Calcium Oscillations in Pancreatic Acini

When the freshly isolated rat pancreatic acini were exposed to tandem doses of ACh (30 nM) or CCK (20 pM), reproducible calcium oscillations were observed (Figure 1a,e). However, if mixed histones (Hx, 50, 150, 200 mgL^−1^, for 30 min) were added in between the tandem doses of ACh or CCK, calcium oscillations induced by the second dose of ACh or CCK were inhibited, dependent on the dose of the mixed histones applied (ACh, Figure 1a–d; CCK, Figure 1e–h). Inhibition of the second ACh (30 nM) stimulation was significant at Hx doses of 150, 200 mgL^−1^ (*P* < 0.05) (Figure 1i). CCK stimulation was more susceptible to Hx inhibition: calcium oscillations induced by the second dose of CCK (20 pM) were inhibited significantly by Hx doses of 50, 150 and 200 mgL^−1^ (*P* < 0.05) (Figure 1j).

Hx inhibition was found to be dependent on the duration of Hx exposure also. The freshly isolated rat pancreatic acini were exposed to Hx 200 mgL^−1^ for 0, 3, 10, and 30 min in between the two tandem doses of ACh (30 nM) (Figure 1k–n) or CCK (20 pM) (Figure 1o–r). Calcium oscillations induced by the second dose of ACh or CCK were inhibited progressively more profoundly with longer Hx exposures. Significant inhibition of calcium oscillations induced by the second dose of ACh (30 nM) (Figure 1s, *P* < 0.05) or CCK 20 pM (Figure 1t, *P* < 0.05) were found with Hx 200 mgL^−1^ at 3, 10 and 30 min, or 10 and 30 min exposure times respectively. These data clearly indicate that histones had no effect on basal calcium in the isolated rat pancreatic acini, but inhibited ACh- or CCK-induced calcium oscillations both concentration- and time-dependently.

### 3.2. Extracellular Histones Induced Robust Calcium Oscillations in AR4-2J Cell Clusters

Rat pancreatic acinar tumor cells AR4-2J retain their major cell surface receptor CCK1: picomolar CCK triggered calcium oscillations which disappeared after the removal of CCK (Figure 2a). In contrast to the lack of effect on basal calcium in freshly isolated rat pancreatic acini (Figure 1), Hx induced robust calcium oscillations in AR4-2J clusters. Hx 2 mgL^−1^ had no apparent effect on basal calcium (Figure 2b), but Hx 5 mgL^−1^ induced strong oscillatory calcium increases, which were even larger than those induced by CCK 20 pM in the same cells (Figure 2c). Higher Hx concentrations (20, 50, 200 mgL^−1^) induced progressively stronger calcium oscillations (Figure 2d–f), which leveled off with an apparent plateau at 50 mgL^−1^ as shown in the dose-response curve (Figure 2g).

To examine the effect of individual histones, recombinant H1, H2A, H2B, H3, and H4 were applied at 0 (a), 2 (b), 5 (c), 20 (d) mgL^−1^ (from left to right in each panel in Figure 3A–E). In each experiment, CCK 20 pM was added first to serve as an internal control; individual recombinant histones were then added after the wash out of CCK. H1, H2A, H2B, H3, and H4 were found to all induce calcium oscillations dose dependently (Figure 3A–E). The minimum effective histone concentration was found to be 2 mgL^−1^ (b); higher concentrations (5, 20 mgL^−1^) (c,d) induced progressively stronger calcium oscillations (Figure 3A–E). Note the varied resting F_340_/F_380_ ratio levels among different cells, which have also been noted before, and are normalized when performing data analysis [46,47,49]. Integrated calcium responses were plotted against histone concentrations for each histone (H1, H2A, H2B, H3, and H4) and are presented in both bar charts and line curves (Figure 3Fa,Fb). These dose-response curves revealed that Hx (data from Figure 2) indeed showed an average efficacy; H2B was the least effective at all concentrations examined (2, 5, 20 mgL^−1^); H2B and H3 were less effective at 2; H2A and H2B were less effective at 5, and H2B was less effective at 20 mgL^−1^ (Figure 3(Fa)). H3 and H4 were more effective than others at 5 mgL^−1^ (Figure 3(Fa)). The dose-response curves illustrated better the efficacy order; from top to bottom it revealed an order of H4 = H3 > Hx = H1 > H2B < H2A, or H4/H3 > H1 > H2A/H2B, with H3 and H4 having the highest efficacies (Figure 3(Fb)). Three distinctive groups of (1) H4/H3, (2) H1 and (3) H2A/H2B were readily recognizable.

### 3.3. Subcellualr Expression of Histone-Targeted TLR2, 4 and 9

RT-PCR showed that both isolated rat pancreatic acini (lanes 1, 2, Figure 4(Aa)) and cultured AR4-2J cells (lanes 3, 4, Figure 4(Aa)) expressed TLR2, TLR4 and TLR9 at levels lower than in pancreatic stellate cells (PSC, lane 5, Figure 4(Aa)) which served as positive controls. The mRNA levels of *TLR2, 9* in pancreatic acini and AR4-2J seemed rather similar, but TLR4 mRNA level was higher in AR4-2J than in pancreatic acini (*P* < 0.05, Figure 4(Ab)). Real-time quantitative PCR (RT-qPCR) was also performed, and it was found that TLR2, 4 and 9 were all expressed but without significant difference in the expression levels between pancreatic acini and AR4-2J cells (data not shown). Western blots revealed that TLR2, TLR4, TLR9 proteins were present in both isolated rat pancreatic acini and in AR4-2J; the overall content in pancreatic acini and AR4-2J was rather similar (*P* > 0.05, Figure 4(Ba,Bb).

Immunocytochemistry revealed distinctive subcellualr distributions of TLR2, 4 and 9 in both isolated pancreatic acini and in AR4-2J. TLR2 and TLR4 were expressed mainly on the apical plasma membrane (TLR4 less brightly also in nucleus), but TLR9 was concentrated in the region of zymogen granules in the pancreatic acinus (Figure 4(Ca)). Note that although TLR4,9 contents seemed comparable to TLR2 in Western blot, in immunocytochemistry, TLR4,9 did not stain as bright as TLR2 in isolated pancreatic acini. In AR4-2J cell clusters, TLR2 was concentrated in neighboring plasma membrane (equivalent to the apical plasma membrane of a pancreatic acinus), but TLR4 was found exclusively in the nucleus (Figure 4(Cb)). TLR9 was densely expressed on the plasma membrane in the periphery of the AR4-2J cell cluster (equivalent to the basal plasma membrane of a pancreatic acinus, Figure 4(Cb)). To identify the TLR subtypes involved in histone induction of calcium oscillations in AR4-2J cells, the effect of TLR2, 4 and 9 ligands was examined.

### 3.4. Extracellular Histone Activation of Peripheral Plasma Membrane TLR9 in AR4-2J Cells

TLR2 agonist zymogen A (10, 100 mgL^−1^) had no effect on basal calcium in AR4-2J cells (Figure 5a,b), nor did TLR4 agonist LPS (1, 10 mgL^−1^) (Figure 5c,d); in both cases, subsequent CCK 20 pM induced robust calcium oscillations (Figure 5a–d). In contrast, TLR9 agonist ODN1826 at 5 μM induced persistent calcium oscillations in AR4-2J, which were rather similar in magnitude to calcium oscillations induced subsequently by CCK 20 pM in the same cells (Figure 5e), and similar in duration to calcium oscillations induced by H4 at 5 mgL^−1^ (Figure 5f). Further, H4-induced calcium oscillations were almost completely blocked by the simultaneous perfusion of TLR9 antagonist ODN2088 (5 μM) (compare Figure 5f,g). ODN2088 blockade of H4-induced calcium oscillations were nearly (88%) complete (Figure 5h, *P* < 0.05). The above data clearly indicate a predominant role of TLR9 in histone-induced calcium oscillations in AR4-2J cells. The concentrations of zymogen A, LPS, ODN1826, and ODN2088 used here were based on previous reports [32,33,34,35,36,37,38].

The peripheral plasma membrane-localized TLR9 in AR4-2J cell clusters are readily accessible to extracellular histones. However, will TLR9 still be activated by extracellular histones if TLR9 migrates to the cell interiors? As shown below, AR4-2J cell culture in dexamethasone (Dex)-containing medium did exactly the trick of internalizing TLR9.

Glucocorticoid Dex has been shown to differentiate AR4-2J cells to the more exocrine cell-like phenotype [50,51]. AR4-2J cells were in the present work exposed to Dex 10 nM in culture for 5 days. Immunocytochemistry confirmed TLR9 expression on peripheral plasma membrane in control AR4-2J cell clusters (Figure 6A). After culture of AR4-2J in Dex-containing medium for 5 days, the peripheral plasma membrane TLR9 had migrated to the cell interiors (Figure 6A). The spatial translocation or internalization of TLR9 was clearly delineated by line-scan along dashed straight lines shown in Figure 6A. Line scans revealed that TLR9 was clearly confined to the peripheral plasma membranes in control AR4-2J: Two peaks at the start and end of the fluorescence curve in Figure 6A. In contrast, TLR9 was mainly present in the cell interiors away from the plasma membrane in AR4-2J cells cultured in Dex-containing medium (Figure 6A): The distinctive peaks in the middle of the fluorescence curve indicate intracellular organelles. The above data clearly indicate that after culture in Dex-containing medium, TLR9 in AR4-2J cells migrated from the peripheral plasma membrane to the cell interiors.

Consistent with immunocytochemical data, it was found that in control AR4-2J cells not exposed to dexamethasone, H4 at 5 mgL^−1^-induced calcium oscillations were 4.65-fold as robust as those induced by CCK 20 pM (D0, Figure 6(Ba,Bf)). After 1 day culture in Dex-containing medium, the strength of H4-induced calcium oscillations were reduced to 2.57-fold of CCK response (D1, Figure 6(Bb,Bf)). The H4 effect was almost completely gone after 3 days of Dex treatment (0.64-fold of CCK response, D3, Figure 6(Bc,Bf)). Longer treatments (for 5 and 7 days respectively) with Dex 10 nM seemed to have resulted in a small rebound effect, but still H4-induced calcium oscillations were smaller than the CCK effect (0.78 and 0.64-fold of CCK response respectively) (D5, D7, Figure 6(Bd–Bf)). Figure 6(Bf) presents the H4/CCK ratios (integrated calcium peaks above baseline) against the duration of Dex treatment; the statistically significant difference from controls was conspicuous (*, *P* < 0.05).

To further confirm a predominant role of TLR9 activation in histone-induced calcium oscillations in AR4-2J, TLR9 expression was suppressed by transfection with TLR9-specific siRNA. In time-matched control AR4-2J cells not transfected with any siRNA, H4 (5 mgL^−1^) induced strong calcium oscillations (Figure 7a). After transfection with TLR9-specific siRNA, H4 (5 mgL^−1^)-induced calcium oscillations in AR4-2J cells were almost completely obliterated (Figure 7b), whereas H4-induced calcium oscillations were little affected in AR4-2J cells transfected with scrambled siRNA (*NC*, Figure 7c). Figure 7d clearly demonstrates that the presence of TLR9 was absolutely required for H4-induced calcium oscillations to occur. Western blot confirmed that TLR9 protein content was markedly reduced after transfection of TLR9-specific siRNA (Figure 7e,f, *P* < 0.05), but TLR9 protein was not significantly altered after treatment with non-specific siRNA (Figure 7e,f, *P* > 0.05). These data further corroborate an essential role of TLR9 in histone-induced calcium oscillations in AR4-2J cells.

Other than direct triggering of calcium oscillations, extracellular histones were also found to sensitize CCK1 receptor activation in AR4-2J cells. Extracellular histone sensitization of CCK1 receptor activation was observed when CCK1 receptors were activated either reversibly by CCK or irreversibly by photodynamic action (see Figure A1).

## 4. Discussion

In the present work we have shown that extracellular histones activated peripheral plasma membrane TLR9 to trigger immediate cytosolic calcium oscillations in rat pancreatic acinar tumor cell AR4-2J. Data to support TLR9 activation as being essential for extracellular histone triggering of calcium oscillations can be divided into the following six independent lines. (1) Extracellular histones had no effect on cytosolic calcium in the freshly isolated rat pancreatic acini, because histone-targeted TLR2 and 4 were present on apical plasma membrane and histone-targeted TLR9 was localized in the zymogen granule regions; therefore, TLR2, 4 and 9 were not readily accessible to extracellular histones. Mixed histones (Hx) were able to block calcium oscillations induced by CCK or ACh possibly due to direct physicochemical interactions between histones and basal plasma membrane. (2) Mixed and individual histones H1, H2A, H2B, H3, and H4 all induced calcium oscillations in AR4-2J cell clusters, because TLR9 was exclusively expressed on the peripheral plasma membrane. The efficacy order to trigger calcium oscillations (H4/H3 > H1/Hx > H2A/H2B) indicates that histone cationic charge density was not important for TLR activation, because the order of isoelectric point was different: H4 > H3 > H2A > H1 > H2B [11]. (3) Neither TLR2 agonist zymosan A nor TLR4 agonist LPS had any effect on basal calcium in AR4-2J cell clusters because TLR2 and 4 were localized on the lateral (neighboring) plasma membrane and in the nucleus respectively—not readily accessible to extracellular histones. (4) TLR9 agonist ODN1826 induced robust calcium oscillations because TLR9 was present on the peripheral plasma membrane in AR4-2J cell clusters. H4-induced calcium oscillations were completely inhibited by TLR9 antagonist ODN2088 in AR4-2J cells. (5) Histones (H4) no longer induced calcium oscillations in AR4-2J cells after treatment with Dex, because after Dex treatment the peripheral plasma membrane TLR9 was internalized. (6) Histone H4 no longer induced calcium oscillations in AR4-2J cells after siRNA knock-down of TLR9 protein expression. The above six independent lines of evidence clearly indicate that extracellular histones activate peripheral plasma membrane TLR9 to trigger calcium oscillations in rat pancreatic acinar tumor cell AR4-2J.

The physiological or pathophysiological relevance of an extracellular histone-mediated blockade of CCK- and ACh-induced calcium oscillations in isolated rat pancreatic acini, and of extracellular histone-triggered calcium oscillations in cultured AR4-2J, observed in the present work, may be corroborated by circulating concentrations of histones found in patients with severe acute pancreatitis, sepsis, sterile inflammation or trauma, as mentioned before [15,16,18,20,21,22]. In taurocholate infusion-induced acute pancreatitis in mice, circulating histones could increase to 149.6 mgL^−1^ [20]. In caerulein injection-induced mouse acute pancreatitis, H3 levels were significantly elevated in central venous blood [23]. Circulating histones increased to 5.9–33.8 mgL^−1^ in patients with severe acute pancreatitis [22]. The relevance of circulating histone concentrations for the modulation of pancreatic acinar cell function is further corroborated by the finding that FITC-labeled histones after tail vein injection could home-in to mouse pancreas [23].

ACh- or CCK-induced calcium oscillations were blocked by lower histone concentrations (50 mgL^−1^) over longer exposure time (30 min) (Figure 1a-j) or by higher histone concentrations (200 mgL^−1^) rapidly (3 min) (Figure 1k-t) in the freshly isolated rat pancreatic acini. Histone inhibition of ACh- or CCK-induced calcium oscillations might well be one of the reasons as to why ACh- or CCK-induced calcium oscillations were blocked by activated neutrophils [6]. In the present work, histones were added to rat pancreatic acini for 30 min maximum (Figure 1). Longer incubation (60 min) with mixed histones (200 mgL^−1^) has been found to lead to necrosis of the isolated mouse pancreatic acini [23]. In any case, histone inhibition of CCK or ACh activation of isolated rat pancreatic acini may at least partially explain the secretory blockade widely reported in acute pancreatitis [6].

Extracellular histones triggered calcium oscillations in AR4-2J dose-dependently (Figure 2). Both mixed histones and individual recombinant histones (bacterial recombinant histones, which would lack eukaryotic specific post-translational modifications) were similarly effective (Figure 3). The efficacy order of recombinant histones was H4/H3 > H1/Hx > H2A/H2B, with H3 and H4 being the most effective, H1 was about equivalent in efficacy to mixed histones (Figure 3).

The above order of efficacy might be compared with the order of effectiveness in other cell types. H4, 2A and 2B (5 mgL^−1^) were found to be more effective than H3 and H1 in epithelial cells [52]. Endothelial colony-forming cells and human umbilical vein endothelial cells were found to be equally sensitive to H2B, H3 and H4 [53]. H3 and H4 inhibited macrophage ingestion of apoptotic thymocytes, but H1 was without any effect [30]. Lys-rich H2A, H2B, H1 were found to be effective antimicrobial proteins, their IC_50_ against *E. coli* were H2B/3.8 μM < H3/10 μM < H4/12.7 μM [54].

Recombinant histones at 5 mgL^−1^ when converted into molar concentrations show an order of H4/0.445 μM > H2B/0.363 μM > H2A/0.357 μM > H3/0.327 μM > H1/0.241 μM. On the other hand, the iso-electric points show an order of H4 (11.3) > H3 (11.02) > H2A (10.95) > H1 (10.11) > H2B (9.76) [11]. Therefore, the order of histone efficacy to trigger calcium oscillations in AR4-2J does not follow either the order of molar concentrations or the order of iso-electric points. The efficacy order must then be an indication of their affinity for the TLR9 receptor protein whose activation is responsible for triggering calcium oscillations in AR4-2J cell clusters.

The histone-targeted TLR2, 4 and 9 were found to be expressed in both isolated rat pancreatic acini and cultured AR4-2J (Figure 4): apical TLR2,4 and granular region TLR9 in isolated rat pancreatic acini, lateral TLR2, nuclear TLR4 and peripheral TLR9 in AR4-2J. Namely, apical TLR2 and TLR4 in pancreatic acinus have migrated to the neighboring plasma membrane and nucleus respectively in AR4-2J; zymogen granule region-localized TLR9 in pancreatic acinus has moved to the peripheral plasma membrane in AR4-2J cell clusters. Dex differentiation of AR4-2J reversed this pattern to return TLR9 from peripheral plasma membrane to the zymogen granule regions (Figure 6). Different subcellular TLR distributions determined whether histones could, or how to, exert any effect on calcium signaling in normal and tumoral pancreatic acinar cells.

The lateral TLR2 agonist zymosan A and nuclear TLR4 agonist LPS were found to be without any effect on basal calcium, whilst basal/peripheral plasma membrane TLR9 agonist ODN1826 induced strong calcium oscillations in AR4-2J cell clusters (Figure 5). H4-induced calcium oscillations were completely inhibited by TLR9 antagonist ODN2088. Further, after siRNA knock down of TLR9 expression, histones no longer induced any calcium oscillations in AR4-2J (Figure 7). Therefore, it is the activation of peripheral plasma membrane TLR9 that induced calcium oscillations in AR4-2J cell clusters. In the freshly isolated rat pancreatic acini, TLR9 was in the zymogen granule regions where extracellular histone could not gain immediate access to, and, therefore, no effect on basal calcium was found (Figure 1). The apical TLR2 and TLR4 were also unlikely to be readily accessed by extracellular histones, therefore, no effect on basal calcium was found in rat pancreatic acini (Figure 1).

It is probably important that TLR 2, 4 and 9 should have specific subcellular localizations appropriate for their functions. It would be exciting in the future to determine the specific roles of lateral TLR2, nuclear TLR4 in AR4-2J cell clusters, and apical TLR2 and 4 and granular region TLR9 in pancreatic acini. How TLR9 might shuttle between the peripheral plasma membrane and the cell interiors would also be an interesting topic.

It is noteworthy that TLR4 expression was increased in pancreatic acinar cell after acute pancreatitis induced by caerulein injections or by taurocholate infusion [55,56]. LPS treatment and caerulein over-stimulation increased TLR4 content in isolated rat pancreatic acini [57]. TLR9 activation was found to stimulate pancreatic stellate cell proliferation and tumorigenesis [58,59].

Although only isolated rat pancreatic acini and cultured AR4-2J were studied in the present work, it is nonetheless interesting to note that TLR9 has been readily detected in pancreatic ducts after caerulein-induced acute pancreatitis [60]; TLR2, 4 and 9 were all detected in human pancreatic ductal adenocarcinoma; TLR9 was found to be plasma membrane-localized in cancerous regions but was exclusively in the cytosol in adjacent normal pancreatic tissue [37]. Therefore, pancreatic acinar and ductal cells share similar TLR9 subcellular distribution patterns after cancerous transformation.

TLR9 was localized mainly in the zymogen granule regions in the isolated rat pancreatic acini but was present only at the peripheral plasma membrane in AR4-2J cell clusters (Figure 4). Such a polarized distribution was reversed by treatment with Dex. After treatment of AR4-2J with Dex (10 nM), TLR9 migrated from the basal plasma membrane to the cell interiors; accordingly, histones no longer induced any calcium increases after Dex treatment (Figure 6). Peripheral plasma membrane localization of TLR9 was therefore required for histones to induce calcium oscillations in AR4-2J. Histones were without any effect on basal calcium in freshly isolated rat pancreatic acini because TLR9 was localized in the zymogen granule regions where extracellular positively-charged histones could not readily gain access due to the plasma membrane barrier. Such spatial separation between TLR9 and histones would explain the lack of effect of extracellular histones on basal calcium in the isolated pancreatic acini (Figure 1). The molecular details for TLR9 localization at the peripheral plasma membrane and for its transfer to the cell interiors in Dex-treated AR4-2J cell clusters are worthy of further investigation in the future. Newly developed imaging and expression techniques might be useful for such investigations [61].

Phagosomal viral or bacterial un-methylated CpG-DNA is the prototypical agonist for mammalian immune cell TLR9 receptors which are normally localized intracellularly [37]. Extracellular self-DNA non-responsive TLR9 is activated by intracellular non-self-DNA and by endogenous mitochondrial DNA [39,62]. Although histones have been reported before to activate TLR9 in non-parenchymal cells in the liver [19], the present work is the only report of direct extracellular histone activation of plasma membrane TLR9 in tumoral pancreatic acinar cells to trigger calcium oscillations. Cell surface TLR9 is recently reported in neutrophils in systemic inflammatory response syndrome (SIRS) where it is believed to play a protective role against aggressive SIRS, although whether histones would play any role in surface TLR9 activation in such a case is not known [63]. It may be noteworthy that high TLR9 cytosolic expression has been found to be associated with a better prognosis for pancreatic ductal adenocarcinoma [37].

## 5. Conclusions

In conclusion, extracellular histones have no effect on basal calcium but block CCK- or ACh-induced calcium oscillations in the freshly isolated rat pancreatic acini; histones trigger calcium oscillations in cultured rat pancreatic acinar tumor cell AR4-2J by activating peripheral plasma membrane TLR9. The inhibitory effects in pancreatic acini provide a unique explanation for the pancreatic secretory blockade in pancreatic acini that is widely reported in acute pancreatitis. Importantly, the histone stimulatory effects in AR4-2J cell clusters link cell surface TLR9 to the calcium signaling pathway which is distinctively different from the canonical or widely-recognized intracellular TLR9-MyD88/Myddosome signaling pathway (Figure 8). Shuttling the TLR9 receptor between the plasma membrane and the cell interiors may therefore provide a useful therapeutic target in pancreatic pathology or pathology of other organs.

## Figures and Tables

**Figure 1 cells-08-00003-f001:**
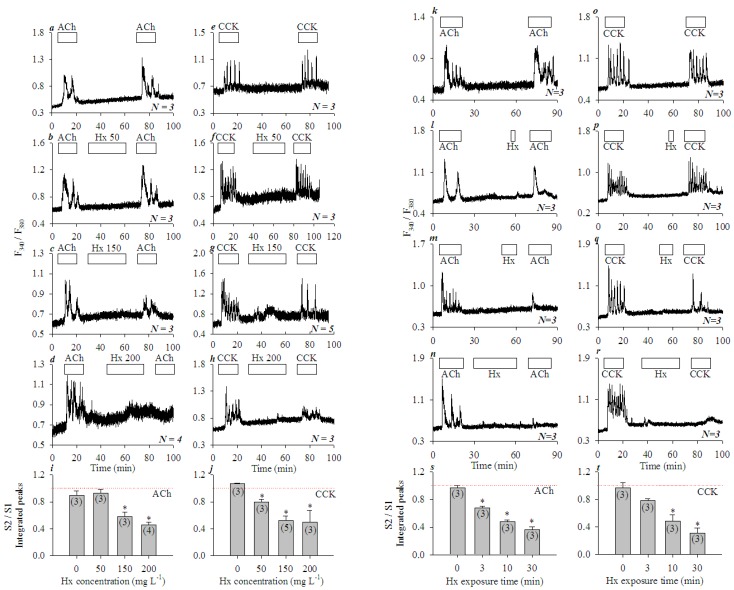
Mixed histones (Hx) inhibited concentration- and time-dependent calcium oscillations induced by ACh or CCK in isolated rat pancreatic acini. Fura-2-loaded rat pancreatic acini were perifused; ACh 30 nM, CCK 20 pM or mixed histones (Hx) were added as indicated by the horizontal bars. Tandem stimulations (S1, S2) of ACh 30 nM (**a**–**d**,**k**–**n**) or CCK 20 pM (**e**–**h**,**o**–**r**) were interspersed with mixed histones (Hx) at 0 (**a**,**e**), 50 (**b**,**f**), 150 (**c**,**g**), 200 (**d**,**h**) mgL^−1^ for 30 min, or at 200 mgL^−1^ for 0 (**k**,**o**), 3 (**l**,**p**), 10 (**m**,**q**), and 30 (**n**,**r**) min. The calcium traces shown are each representative of *N* identical experiments (*N* = 3–5) with pancreatic acini isolated from different rats. For statistical analysis, integrated calcium responses (area under calcium peaks above the basal level) of the 2nd (S2) over the 1st (S1) ACh or CCK stimulations (S2/S1) were plotted against Hx concentrations (**i** from **a**–**d**, **j** from **e**–**h**), or against the duration of Hx (200 mgL^−1^) exposure (**s** from **k**–**n**, **t** from **o**–**r**). Asterisk (*) indicates *P* < 0.05 in comparison with controls (Hx concentration or exposure time zero in **i**,**j**,**s**, and **t** respectively); thin red dashed lines indicate position 1.00.

**Figure 2 cells-08-00003-f002:**
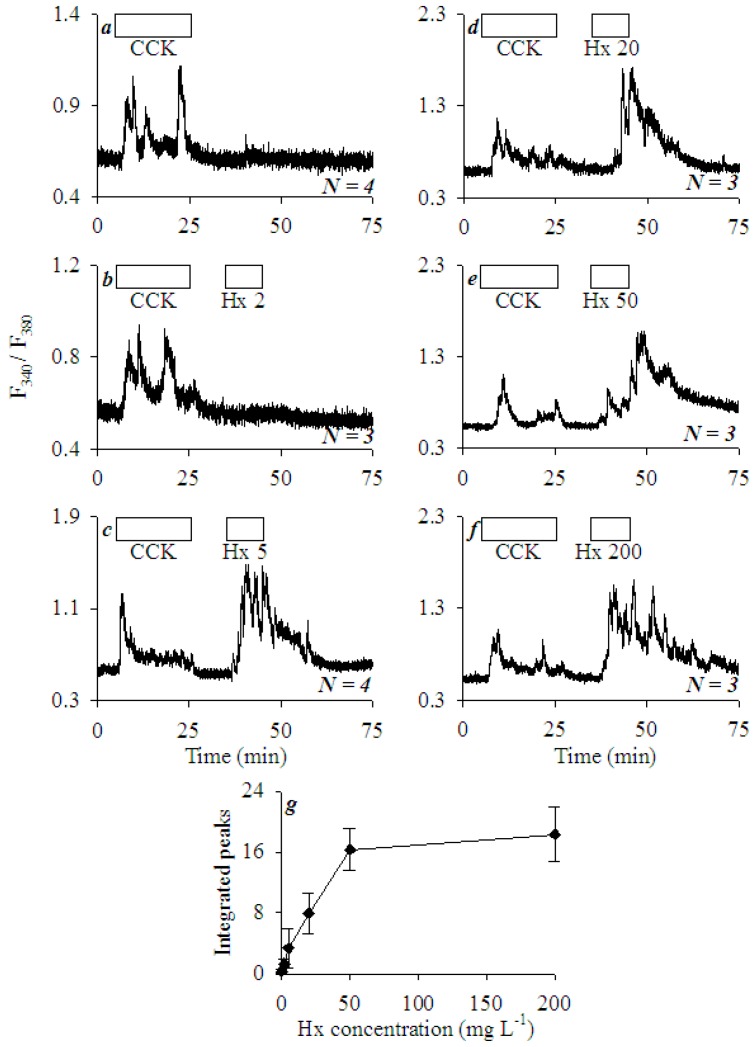
Mixed histones (Hx) induced calcium oscillations in AR4-2J cells. Fura-2-loaded AR4-2J cells were perifused; CCK (20 pM) and mixed histones (0, 2, 5, 20, 50, and 200 mgL^−1^) were added as indicated by the horizontal bars. CCK (20 pM) was added, followed by mixed histones (Hx) at 0 (**a**), 2 (**b**), 5 (**c**), 20 (**d**), 50 (**e**), and 200 (**f**) (mgL^−1^). The calcium traces shown are each representative of *N* identical experiments as indicated. Integrated Hx response (area under calcium peaks above the basal level, 35–75 min) from (**a**–**f**) was plotted against histone concentrations (**g**). Data were expressed as mean ± SEM.

**Figure 3 cells-08-00003-f003:**
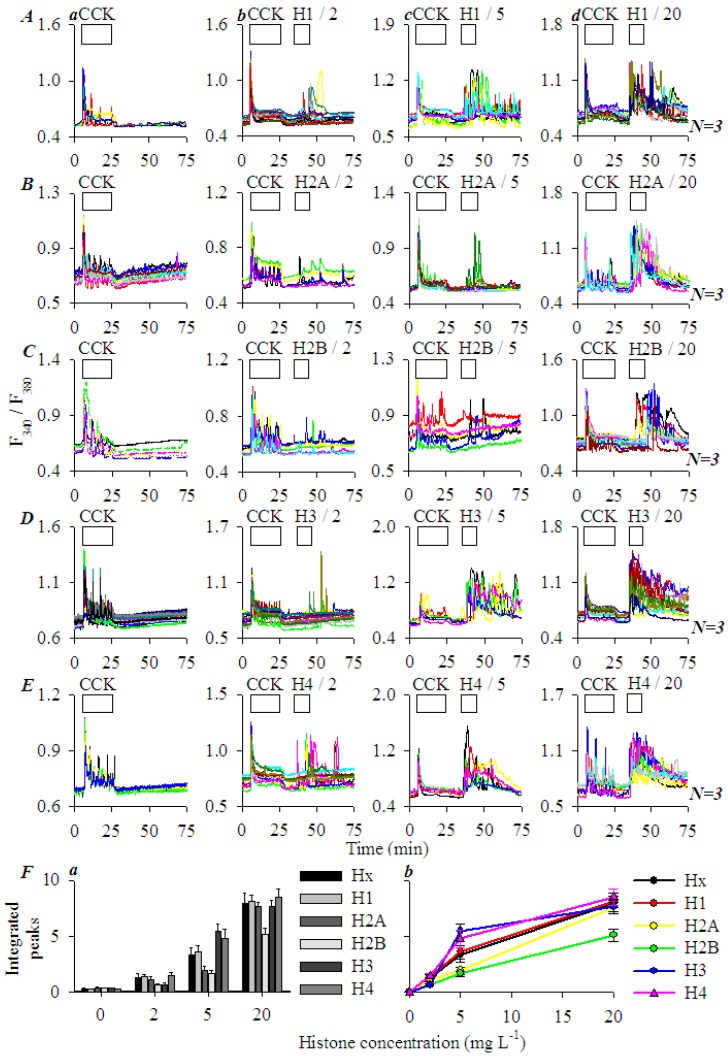
H1, H2A, H2B, H3, and H4 induced calcium oscillations in AR4-2J cells with varied efficacies. Fura-2-loaded AR4-2J were perifused; CCK 20 pM, H1 (**A**), H2A (**B**), H2B (**C**), H3 (**D**), H4 (**E**) at 0 (**a**), 2 (**b**), 5 (**c**), 20 (**d**) mgL^−1^ (from left to right in each panel) were added as indicated by the horizontal bars (*N* = 3). Integrated histone response (area under calcium peaks above the basal level from 35–75 min) was plotted against histone concentrations for each histone, either in bar graphs (**Fa**) or curves (**Fb**).

**Figure 4 cells-08-00003-f004:**
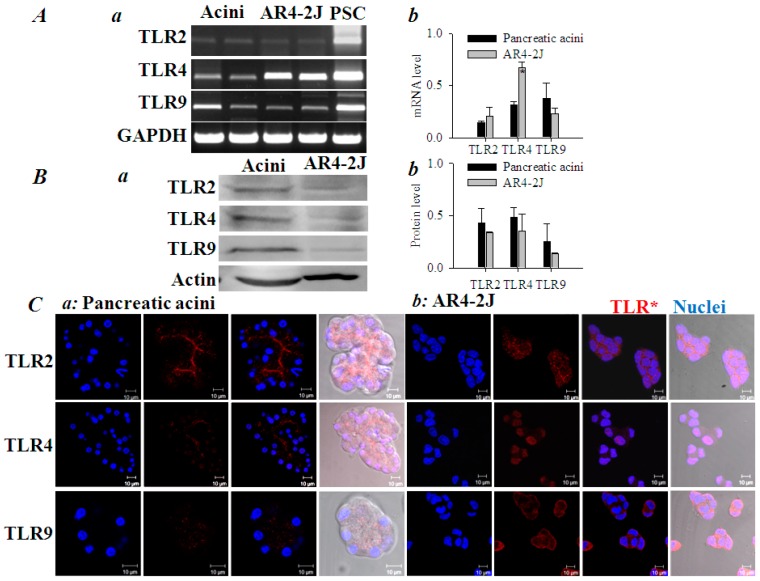
Expression of TLR2, TLR4, TLR9 in pancreatic acini and in AR4-2J cell clusters as determined by RT-PCR, Western blot and immunocytochemistry. (**A**) RT-PCR determination of TLR2, 4 and 9 (**Aa**). Activated pancreatic stellate cells (PSC) were used as positive controls. Lanes for acini or AR4-2J in the images were done with pancreatic acini from two different rats, or from two different AR4-2J cultures. TLR2, 4 and 9 mRNA contents normalized to that of GAPDH were quantified both in rat pancreatic acini and in AR-2J (**Ab**). (**B**) Protein extracts from rat pancreatic acini or AR4-2J were separated by SDS-PAGE, and Western blot was done using antibodies against TLR2, 4 and 9. The β-actin was used as internal controls (**Ba**). TLR2, 4 and 9 and β-actin bands in blots from ≥ 3 rats or ≥ 3 AR4-2J cultures were quantified by ImageJ; TLR2, 4 and 9 protein levels were expressed as ratios of TLR/β-actin. No statistically significant differences were found (*P* > 0.05, **Bb**). (**C**) Rat pancreatic acini or AR4-2J were fixed for immunocytochemistry. Antibodies against TLR2, 4 and 9 and TRITC-conjugated secondary antibody were used (red). The nucleus was counter-stained with Hoechst 33342 (blue). Confocal images were taken (Zeiss 700) with λ_ex_ 555 nm (TRITC) or 405 nm (Hoechst 33342), λ_em_ 576 nm and 461 nm respectively. Images from one typical experiment are shown. Scale bars: 10 μm. Both fluorescent and merged images are presented.

**Figure 5 cells-08-00003-f005:**
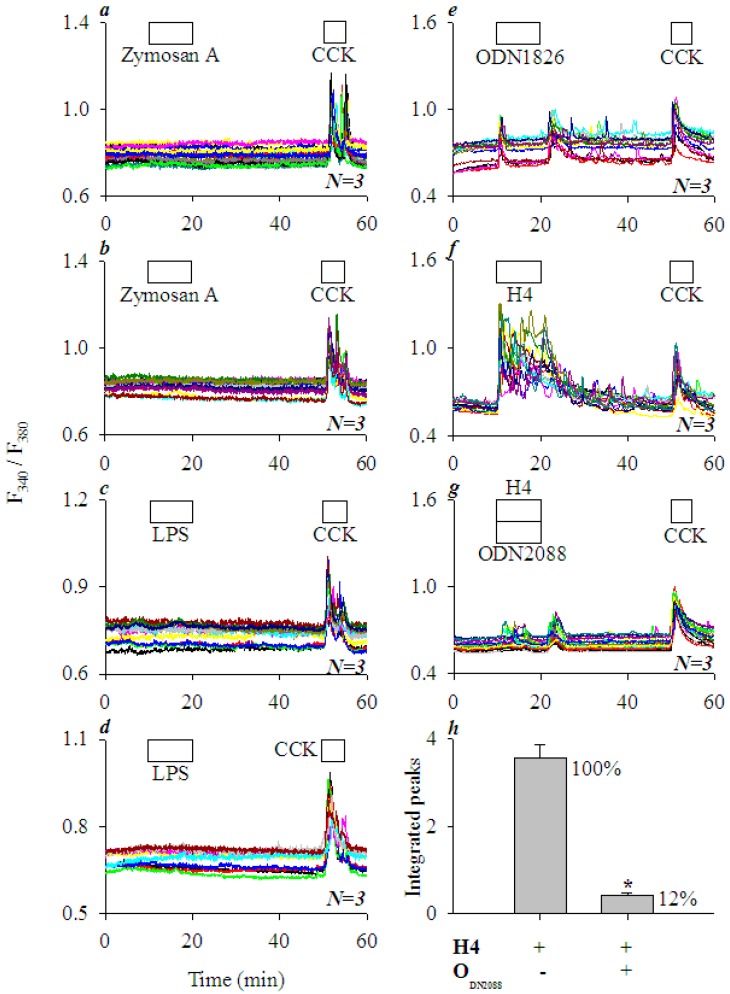
TLR2 and TLR4 agonists had little effect but TLR9 agonist elicited calcium oscillations; histone-induced calcium oscillations were blocked by TLR9 antagonist in AR4-2J cells. Fura-2-loaded AR4-2J were perifused; TLR2 agonist zymosan A, TLR4 agonist LPS, TLR9 agonist ODN1826, CCK, H4, or TLR9 antagonist ODN2088 were added as indicated by the horizontal bars. TLR2 agonist zymosan A (10, 100 mgL^−1^, **a**,**b**), TLR4 agonist LPS (1, 10 mgL^−1^, **c**,**d**) and TLR9 agonist ODN1826 (5 μM, **e**) were added before CCK (20 pM) (**a**–**e**). H4 (5 mgL^−1^, **f**) alone or H4 (5 mgL^−1^) plus TLR9 antagonist ODN2088 (5 μM, **g**) were added before CCK (20 pM, **f**,**g**). Note that H4-induced calcium oscillations were blocked by simultaneous addition of TLR9 antagonist ODN2088 (**g**), as confirmed by statistical analysis (area under the peaks above the basal level, **h**). The asterisk (*) in (**h**) indicates *P* < 0.05. The calcium traces shown in (**a**–**g**) are each representative of *N* identical experiments (*N* = 3).

**Figure 6 cells-08-00003-f006:**
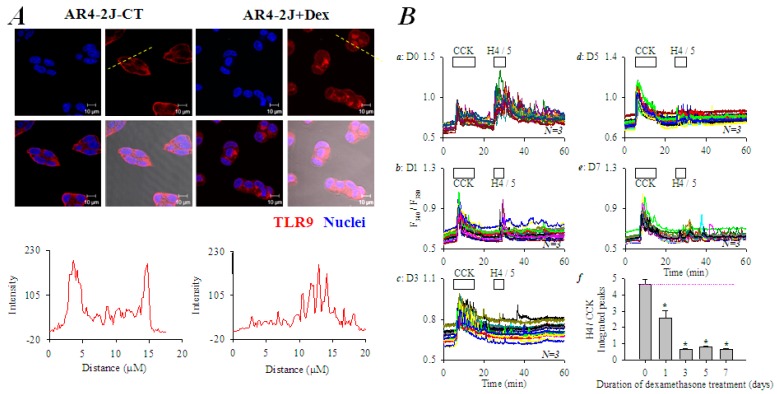
TLR9 migrates from the peripheral plasma membrane to cell interiors after Dex treatment, with concurrent disappearance of histone-induced calcium oscillations in AR4-2J cell clusters. (**A**) TLR9 migration from peripheral plasma membrane to cell interiors after Dex. AR4-2J cells were grown in normal medium (AR4-2J-CT) or in medium containing Dex 10 nM (AR4-2J + Dex) for 5 days before immunocytochemistry. Anti-TLR9 primary and TRITC-conjugated secondary antibodies were used (red). The nucleus was counter-stained with Hoechst 33342 (blue). Confocal images were taken (Zeiss 700) with λ_ex_: TRITC/555 nm, Hoechst 33342/405; λ_em_: 576 nm and 461 nm respectively. Both fluorescent and merged images are shown. The distribution of TLR9 protein content along the dashed thin yellow lines in the images was quantified by fluorescence line scans. Fluorescence intensity was concentrated on the plasma membrane and cell interiors in control (left) and Dex-treated (right) AR4-2J cells respectively. (**B**) Disappearance of histone-induced calcium oscillations after Dex. AR4-2J cells were treated with Dex 10 nM for 0, 1, 3, 5, 7 days (**a**–**e**) before detachment and measurements of H4 induction of calcium oscillations. Fura-2-loaded AR4-2J cells were perifused; CCK and H4 were added as indicated by the horizontal bars. In each experiment, CCK 20 pM was followed by H4 at 5 mgL^−1^ in AR4-2J treated with Dex 10 nM for 0 (**a**), 1 (**b**), 3 (**c**), 5 (**d**) or 7 days (**e**). Ratios of integrated calcium responses H4 (25–55 min)/CCK (5–15 min) (area under peaks above the basal level) were plotted against days of Dex treatment (**f**). The dashed thin pink horizontal line in (**f**) indicates the ratio (H4/CCK) in control AR4-2J not treated with Dex. Asterisk (*) indicates *P* < 0.05.

**Figure 7 cells-08-00003-f007:**
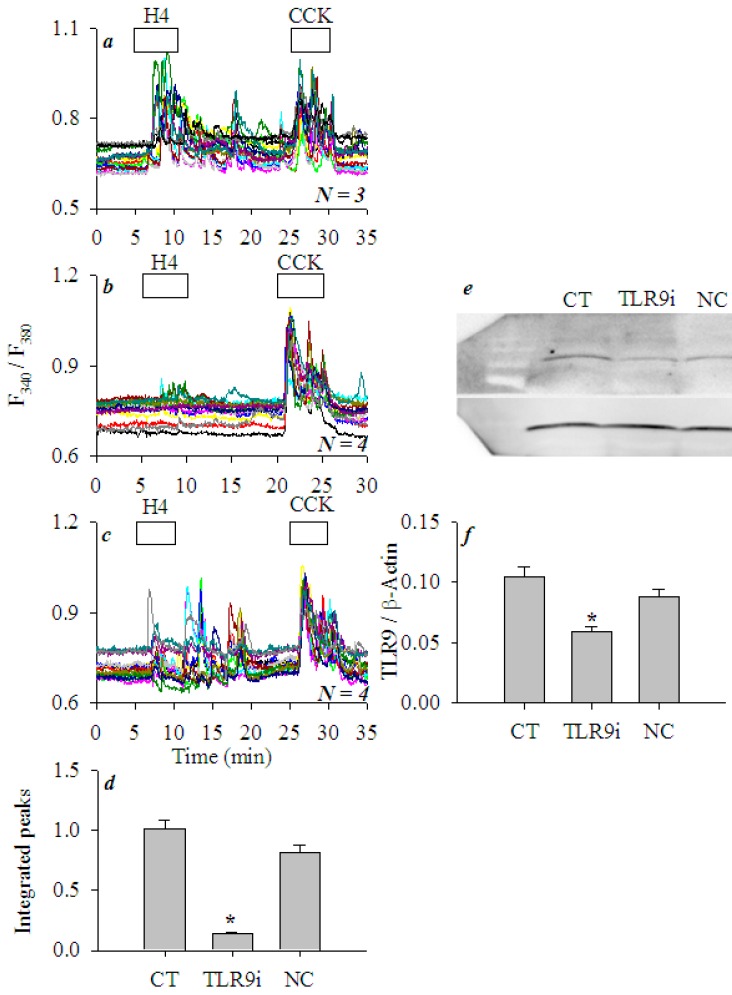
siRNA down-regulation of TLR9 expression blocked histone-triggered calcium oscillations in AR4-2J cells. AR4-2J cells were cultured as normal control (**a**), transfected with siRNA-TLR9 (**b**) or with scrambled siRNA (**c**), loaded with Fura-2 AM, and perifused. H4 (5 mgL^−1^) and CCK (20 pM) were added as indicated by the horizontal bars. The H4-induced calcium oscillations (**a**–**c**, peak area above baseline from 5–20 min) were calculated (**d**). Western blot images (**e**) and statistical analysis (**f**, *N* = 3) of TLR9 protein content are presented. Abbreviations: CT, not transfected; TLR9i, transfected with siRNA-TLR9; *NC*, transfected with scrambled sequence. AR4-2J cells were analyzed for both calcium and TLR9 protein content 24 h after transfection. Protein content of -actin was used as an internal reference for TLR9 (TLR9/-actin). Asterisk (*) indicates statistical significance at *P* < 0.05.

**Figure 8 cells-08-00003-f008:**
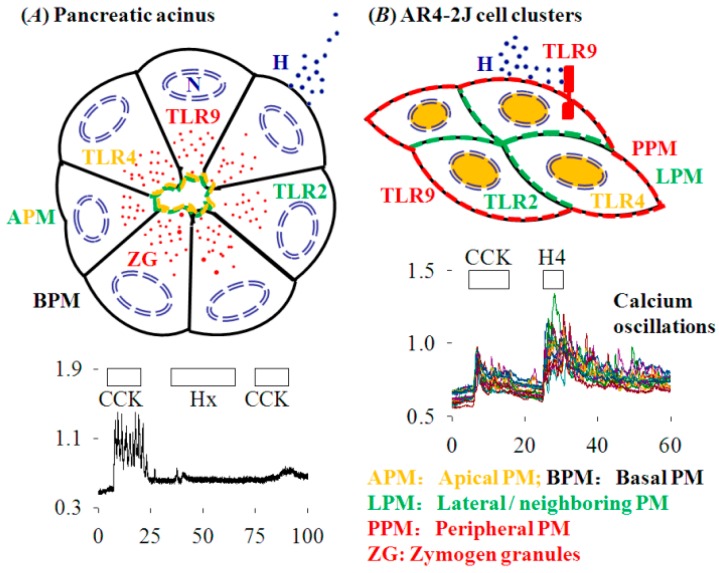
Effects of extracellular histones on calcium signal in rat normal and tumoral pancreatic acinar cells. (**A**) Histones have no effects on basal calcium but block CCK stimulation in rat pancreatic acinus. Each acinar cell has one apical, two lateral and one basal side. The lower panel indicates that in tandem stimulations the second dose of CCK does not induce calcium oscillations after histone treatment (Hx). (**B**) Histones stimulate TLR9 to elicit calcium oscillations in AR4-2J cell clusters. Each cell has two neighboring and one basal/peripheral side. The lower panel shows that H4 induces calcium oscillations which are stronger and longer-lasting than CCK stimulation. The color-coded abbreviations are listed. APM, apical plasma membrane; BPM, basal plasma membrane; CCK, cholecystokinin; H, histones; Hx, mixed histones; LPM, lateral/neighboring plasma membrane; N, nucleus; PPM, peripheral plasma membrane; TLR2, 4 and 9, Toll-like receptor 2, 4 and 9; ZG, zymogen granules. The calcium traces shown in (**A**,**B**) lower parts are taken from Figure 1r and Figure 6(Ba) respectively. For clarity, CCK1 receptors are omitted from the diagram.

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
