# Peer review of "Extracellular Histones Activate Plasma Membrane Toll-Like Receptor 9 to Trigger Calcium Oscillations in Rat Pancreatic Acinar Tumor Cell AR4-2J"

_cells, 2018, doi:10.3390/cells8010003_

Round 1

Reviewer 1 Report

Summary

Guo and co-workers study the effects of histones on the generation of Ca2+ oscillations in rat pancreatic acini and AR4-2J tumor cells. They find that histone treatment of the primary cells leads inhibits Ca2+ oscillations generated by CCK or ACh. In contrast, they find that histones can elicit Ca2+ oscillations in the AR4-2J. They attribute these differences to the different spatial localization of TLR2, TLR4 and TLR9 receptors, where AR4-2J cells have TLR9 localized on the peripheral plasma membrane. Overall, the work is important and makes a strong case for the TLR9-mediated oscillations observed in the AR4-2J cells. However, there is no doubt that additional experiments could strengthen the work further and make the paper more sound. My specific major and minor concerns are listed below:

Major

1.       Figure 1 – Why is the resting Fura ratio (prior to the addition of ACh or CCK) different in panels A and B compared to C and D? Doesn’t this difference imply an effect on basal Ca2+? Some explanation is needed.

2.       Lines 195-196 – “basal calcium” implies cytosolic Ca2+ levels prior to cell stimulation while the cells are at rest. The authors should appropriately define this term in the paper and ensure they are using it in the correct context.

3.       Figure 3 – Some of the panels show large variations in the resting Ca2+ levels which could have an effect on the amplitude and frequency of the oscillations which are produced (see for example panel C). Some discussion and explanation is warranted.

4.       Figure 4A – Quantitative RT PCR will give a better estimation of relative expression levels. The authors should consider performing these experiments which should be straight forward since the primers are already designed and are known to work.

5.       Figure 4B – TLR protein levels look much lower in AR4-2J cells compared to acini, but the quantification suggests that there is no significant difference. Are these the best representative blots?

6.       Figure 4C – The acini TLR staining for TLR4 and TLR9 is hardly noticeable, yet in the panel B the western suggests TLR should be present. Further, the AR4-2J cells seem to show higher levels of TLRs than acini, yet the western blots suggest the opposite. Some explanation for the reasons for these discrepancies is required.

7.       Figure 5 – What is the basis for the Zymosan A, LPS, etc. concentrations used in this set of experiments? References or rationale are needed to using these concentrations.

8.       Figure 6 – The only experimental evidence for the TLR9 redistribution is the fluorescence line scans. Another method (i.e. fractionation followed by western) would strengthen this conclusion.

9.       Figure 7 – The western shows that TLR9 is still present after siRNA treatment. Why not show RT-qPCR data? Also, the western is not convincing since only a sliver of the band is shown. The full gel/blot should be shown.

10.   Lines 362-364: This data is not well integrated into the rest of the manuscript. It appears that it was added just as an afterthought. If this data remains in the manuscript, some rationale as to why these experiments were performed should be included and how they fit into the TLR9 receptor aspect of the manuscript.

11.   The authors discuss the isoelectric point of histones extensively. A reference or references should be provided showing that histone interactions with TLR9 (or other TLRs) is charge dependent.

12.   How can histones inhibit in acini but activate Ca2+ oscillations in AR4-2J cells but inhibit in primary acini? The signaling pathways in each of these scenarios should be included as a final model figure including specifically the underlying proteins involved in generating the Ca2+ oscillations themselves. This figure should also include the relative locations of the TLRs studied.

Minor

1.       Ensure that all acronyms are defined on first usage.

2.       Manuscript requires a careful review of grammar and language.

3.       I would suggest removing the claims of being the “first” report to demonstrate TLR9 activation (and calcium oscillations) via histones.

4.       Subheadings are too long.

5.       Figure 3 resolution is low, making it difficult to carefully examine the data.

6.       Figure 4 is too small – the text is difficult to read making the data hard to interpret. There seems to be multiple panel labels referring to the same panel (i.e. A versus a). This makes the figure disorganized and confusing.

7.       In some instances the authors mention the cells are “perfused” and in other instances they state that the cells were “perifused”. What is the difference?

Author Response

Guo and co-workers study the effects of histones on the generation of Ca2+ oscillations in rat pancreatic acini and AR4-2J tumor cells. They find that histone treatment of the primary cells leads inhibits Ca2+ oscillations generated by CCK or ACh. In contrast, they find that histones can elicit Ca2+ oscillations in the AR4-2J. They attribute these differences to the different spatial localization of TLR2, TLR4 and TLR9 receptors, where AR4-2J cells have TLR9 localized on the peripheral plasma membrane. Overall, the work is important and makes a strong case for the TLR9-mediated oscillations observed in the AR4-2J cells. However, there is no doubt that additional experiments could strengthen the work further and make the paper more sound. My specific major and minor concerns are listed below:

Major

1.       Figure 1 – Why is the resting Fura ratio (prior to the addition of ACh or CCK) different in panels A and B compared to C and D? Doesn’t this difference imply an effect on basal Ca2+? Some explanation is needed.

The resting ratios F340 / F380 do vary among different cells, even when imaging in the same light field. For Fig. 1, when doing quantitative analysis the integrated peaks due to the second stimulation of ACh or CCK (S2) were divided by (i.e., normalized to) the integrated peaks triggered by the first stimulation with ACh or CCK (S1), therefore when analyzing S2 / S1 ratios one need not worry too much about such variations.

In all the experiments presented in this paper, we were mainly concerned with temporal changes in calcium concentrations. We did not observe any difference in cellular responses due to varied initial ratios of F340 / F380. The variation could be due to factors not related to cellular function, for example due to slight variations among different cells in Fura-2 loading level. Baseline calcium concentration is believed not to vary much among healthy cells.

Varied resting ratios of F340 / F380 are very normal among any cohort of cells or among cells from different experiments. In Section 2.7, Para 1 Line 8: we have added the sentence "Note the varied resting calcium levels among different cells." Modifications in text are all marked in red.

2.       Lines 195-196 – “basal calcium” implies cytosolic Ca2+ levels prior to cell stimulation while the cells are at rest. The authors should appropriately define this term in the paper and ensure they are using it in the correct context.

The phrase “basal calcium” first appeared in Section 1 Last Para Lines 3-4: where immediately after the word "basal", we have now added a bracket: “(i.e., resting)”.

3.       Figure 3 – Some of the panels show large variations in the resting Ca2+ levels which could have an effect on the amplitude and frequency of the oscillations which are produced (see for example panel C). Some discussion and explanation is warranted.

As mentioned in point 1 above, varied resting calcium levels did not seem to affect subsequent stimulated responses. This also did not affect our quantitative analysis of calcium responses.

For statistical analysis in all figures involving calcium responses (Figs. 1-3, 5-7, A1), the integrated peaks is defined as: calcium peak values (F340 / F380 ratios) for each data point in a certain time period (35 - 75 min in Fig. 3, for example) minus basal calcium level, then the difference for each data point was added together to obtain the integrated peaks. For Fig. 1, the integrated peaks of S2 were also divided by the integrated peaks of S1.

As a result, we have added in text in Section 3.2, Para 2 Lines 8 the following: "Note the varied resting calcium levels among different cells."

Also in the relevant figure legends (Figs. 1-3, 5, 6, A1), “area under calcium peaks” has been changed to “area under calcium peaks above the basal level”.

4.       Figure 4A – Quantitative RT PCR will give a better estimation of relative expression levels. The authors should consider performing these experiments which should be straight forward since the primers are already designed and are known to work.

Real time quantitative PCR is now done and data confirmed TLR2,4,9 expression as shown by the semi-quantitative RT-PCR. Fig. 4A was to confirm that TLR2,4,9 were expressed at the mRNA level in both isolated acini and AR4-2J, strict quantification or comparison of their quantities were not essential.

Data from one typical experiment and the average of three experiments are as follows.

Section 3.3 Para 1 Line 5: In text we have added “Real time quantitative PCR (RT-qPCR) confirmed these expression patterns (data not shown)”.

Also after re-checking our original data from the semi-quantitative RT-PCR, we have now replaced Fig. 4Ab, but the general pattern remained the same with only very slight changes in the mean / SEM.

5.       Figure 4B – TLR protein levels look much lower in AR4-2J cells compared to acini, but the quantification suggests that there is no significant difference. Are these the best representative blots?

These are the better ones. There were variations among experiments (blots), especially with pancreatic acini from different rats.

6.       Figure 4C – The acini TLR staining for TLR4 and TLR9 is hardly noticeable, yet in the panel B the western suggests TLR should be present. Further, the AR4-2J cells seem to show higher levels of TLRs than acini, yet the western blots suggest the opposite. Some explanation for the reasons for these discrepancies is required.

The quantitative respect of immunocytochemistry (ICC) seemed at variance with Western blot. In Section 3.3 Para 2 Line 4, we have added a new sentence to address this point: "Note that although TLR4,9 contents seemed comparable to TLR2 in Western blot, in immunocytochemistry TLR4,9 did not stain as bright as TLR2 in isolated pancreatic acini. " Multiple experiments were done for TLR4,9 staining in pancreatic acini, including time-matched experiments together with TLR2 staining. TLR2 in pancreatic acini stained better in immunocytochemistry, as TLR2,4,9 did in AR4-2J.

7.       Figure 5 – What is the basis for the Zymosan A, LPS, etc. concentrations used in this set of experiments? References or rationale are needed to using these concentrations.

Most such references were quoted in Introduction, although the detailed concentration ranges were omitted for clarity. We have now added a new sentence at the end of Para 1 in Section 3.4: "The concentrations of zymogen A, LPS, ODN1826, ODN2088 used here were based on previous reports [32-38]. " 

8.       Figure 6 – The only experimental evidence for the TLR9 redistribution is the fluorescence line scans. Another method (i.e. fractionation followed by western) would strengthen this conclusion.

Cell fractination could be used to isolate the plasma membrane and intracellular TLR9 proteins to examine whether there is any difference in the TLR9 protein molecules, or whether there are any interactions of TLR9 with other transmembrane proteins. For subcellular localization and subcellular transfer of proteins, immunocytochemistry is well accepted. The shuttling of TLR9 protein between peripheral plasma membrane and cell interiors is an important topic, we are likely to pursue this topic further in the future. For transmembrane proteins (intracellular TLR9 is also known to be a transmembrane protein), very stringent conditions may be needed to separate the fractions of plasma membrane and intracellular membranes (ER membranes, for example). The small amount of PM membranes may be contaminated by the larger ER membranes. The task would be easier if the protein tranfers only between cytosol and membrane, or between cytoplasma and nucleus.

9.       Figure 7 – The western shows that TLR9 is still present after siRNA treatment. Why not show RT-qPCR data? Also, the western is not convincing since only a sliver of the band is shown. The full gel/blot should be shown.

We sometimes do immunocytochemistry after siRNA knock-down, but Western is quantitative and quantitative for protein content (not for mRNA content). We have now redone the image so that areas both below and above the TLR9 band is shown. The new panel (e) in Fig. 7 is now like the following:

10.   Lines 362-364: This data is not well integrated into the rest of the manuscript. It appears that it was added just as an afterthought. If this data remains in the manuscript, some rationale as to why these experiments were performed should be included and how they fit into the TLR9 receptor aspect of the manuscript.

In my version of the MS downloaded from the Cells Website, Lines 362-364 are in the Discussion - Section 4 Para 1 Lines 13-15. These lines discuss the histone efficacy order (not relevant to isoelectric point order – this latter order is from Ref 11), and TLR2,4 receptor pharmacology. These are the main data of the MS.

If the Referee is referring to the last two lines in Results (description of data presented in Appendix A), we have the following reply.

We have modified the two lines to make them more succinct and clear. The new passage is as follows: “Other than direct triggering of calcium oscillations, extracellular histones were also found to sensitize CCK1 receptor activation in AR4-2J cells. Extracellular histone-sensitization of CCK1 receptor activation was observed when CCK1 receptors were activated either reversibly by CCK or irreversibly by photodynamic action (see Fig. A1).

These data are an extension of the data presented in the main body of the MS, therefore they are only presented as an Appendix A. Histones not only inhibit intact acini and activate TLR9 in AR4-2J, they also sensitize both CCK and photodynamic activation of CCK1 receptors in AR4-2J. This is an extra characterisctics of the hisotone effect in pancreatic acinar cells AR4-2J. Appendix A / Fig. A1 are an extension of the main text. Removal of Appendix A will not affect the conclusion of the main text but it enriches the main text. 

11.   The authors discuss the isoelectric point of histones extensively. A reference or references should be provided showing that histone interactions with TLR9 (or other TLRs) is charge dependent.

Histone efficacy order did not follow their isoelectric point order, therefore histone charge density was not responsible for histone activation of TLR9 and triggering of calcium oscillations in AR4-2J. Such cationic charges are important for their DNA binding capacity. We do not know that histone activation of TLR2,4,9 is charge dependent. It is just a discussion point which is ruled out.

12.   How can histones inhibit in acini but activate Ca2+ oscillations in AR4-2J cells but inhibit in primary acini? The signaling pathways in each of these scenarios should be included as a final model figure including specifically the underlying proteins involved in generating the Ca2+ oscillations themselves. This figure should also include the relative locations of the TLRs studied.

Extracellular mixed histones may exert their inhibitory effects by direct physicochemical interactions with the plasma membrane microenvironment (see Discussion Para 1 Line 9) in pancreatic acini (acini being more susceptible than the tumoral AR4-2J cells?) so that the microenvironment deviates from optimal conditions for cell surface receptor activation. But we are reluctant to speculate before more data are obtained, possibly in the near future.

We have now included a new Fig. 8 in Discussion to summarize all the data presented in this MS. Since how TLR9 activation results in calcium oscillations is not yet known, Fig. 8 outlines only available data.

Minor

1.       Ensure that all acronyms are defined on first usage.

We have done so as advised. Changes are marked in red in the revised MS.

2.       Manuscript requires a careful review of grammar and language.

We have now revised the text further throughout. All changes are marked in red in the revised version of the MS.

3.       I would suggest removing the claims of being the “first” report to demonstrate TLR9 activation (and calcium oscillations) via histones.

We have now removed the word “first” and re-phrased the sentence in Section 1, Last Para Line 7 as follows: “To the best of our knowledge histone activation of plasma membrane TLR9 to induce calcium oscillations has not been reported before in any cell type, especially in rat pancreatic acinar cells.

4.       Subheadings are too long.

We have now shortened the subtitles in most cases, especially in Results.

5.       Figure 3 resolution is low, making it difficult to carefully examine the data.

We have enlarged Fig. 3 in the revised version of the MS. All original figures files are now uploaded with re-submission of the revised MS.

6.       Figure 4 is too small – the text is difficult to read making the data hard to interpret. There seems to be multiple panel labels referring to the same panel (i.e. A versus a). This makes the figure disorganized and confusing.

We have enlarged Fig. 4 in the revised MS. Fig. 4 is divided into Panels A, B, C, each panel is further divided into Aa, Ab, etc.

7.       In some instances the authors mention the cells are “perfused” and in other instances they state that the cells were “perifused”. What is the difference?

In this MS these two words were used interchangeably, the publisher may decide their preference or use both simultaneously.

Reviewer 2 Report

Guo et al studied role of histones in acute pancreatitis in via modulating Ca oscillations  by TLR-9 in Rat Acinar Tumor cells. Authors have nicely shown the the induction of calcium oscillation by extracellular histones in AR4-2J cells. They also showed how this can change the subcellular localization of TLRs , specifically TLR9 in plasma membrane to activate Ca oscillations. Study is very nicely done and my comments as follows.

1)The writing style can be improved to make few things clear.

2)Most of the work has been done with transformed rat cells. Authors have not shown if this is true for human pancratic cells as well. If authors can add that, it would definately add some more weight to the paper. Not only that it would add some more information into the field as well.

Author Response

Guo et al studied role of histones in acute pancreatitis in via modulating Ca oscillations by TLR-9 in Rat Acinar Tumor cells. Authors have nicely shown the the induction of calcium oscillation by extracellular histones in AR4-2J cells. They also showed how this can change the subcellular localization of TLRs , specifically TLR9 in plasma membrane to activate Ca oscillations. Study is very nicely done and my comments as follows.

1) The writing style can be improved to make few things clear.

We have now further improved the English usage in the MS. All modifications are marked in red in the revised version of the MS.

2) Most of the work has been done with transformed rat cells. Authors have not shown if this is true for human pancreatic cells as well. If authors can add that, it would definately add some more weight to the paper. Not only that it would add some more information into the field as well.

The title of our MS is “Extracellular histones activate plasma membrane Toll like receptor 9 to trigger calcium oscillations in rat pancreatic acinar tumor cell AR4-2J”. It is always very gratifying if human cells can be secured to further the findings. For pancreatic acinar tumor cell lines, only the rat pancreatic acinar cell line AR4-2J shows regular calcium oscillations without fail whenever the cell surface receptors are activated one way or another. In fact, we have tried to examine the mouse pancreatic acinar cell line MPC-83 which was first reported in 1983 by a group of workers in Kuming, China. Disappointingly, no calcium responses were ever triggered with even very high concentrations of CCK or ACh (Cheng et al., 2008). Some human pancreatic acinar cell lines (such as Panc-1) indeed do show calcium oscillations when stimulated with endogenous agonist (neurotensin) after treatment with insulin, but even after insulin only a higher proportion of cells show calcium oscillations (Young, Rozengurt, 2010). It is known that some human pancreatic samples dissected during cancer operations (normal tissue on the border of tumor) show receptor-mediated calcium oscillations (Ji et al., 2001; Liang et al., 2017; Murphy et al., 2008), but we do not have ready access to such human tissues. Certainly we will be looking for such opportunities to further our work with human tissues in the future.

The AR4-2J cells show regular calcium oscillations, they were chosen to examine histones effects. Calcium oscillations / calcium imaging were used as the real time marker / method to monitor instant changes in cellular activity or function.

References

Cheng HL, Wang L, Cui ZJ (2008) The loss of fucntional G protein-coupled receptors in mouse pancreatic acinar cell line MPC-83. World Chin J Digestol 16: 590-595.

Ji B, Bi Y, Simeone D, Mortensen RM, Logsdon CD (2001) Human pancreatic acinar cells lack functional responses to cholecystokinin and gastrin. Gastroenterology 121: 1380-1390.

Liang T, Dolai S, Xie L, Winter E, Orabi AI, Karimian N, Cosen-Binker LI, Huang YC, Thorn P, Cattral MS, Gaisano HY (2017) Ex vivo human pancreatic slice preparations offer a valuable model for studying pancreatic exocrine biology. J Biol Chem 292: 5957-5969.

Murphy JA, Criddle DN, Sherwood M, Chvanov M, Mukherjee R, McLaughlin E, Booth D, Gerasimenko JV, Raraty MG, Ghaneh P, Neoptolemos JP, Gerasimenko OV, Tepikin AV, Green GM, Reeve JR Jr, Petersen OH, Sutton R (2008) Direct activation of cytosolic Ca2+ signaling and enzyme secretion by cholecystokinin in human pancreatic acinar cells. Gastroenterology 135: 632-641.

Young SH, Rozengurt E (2010) Crosstalk between insulin receptor and G protein-coupled receptor signaling systems leads to Ca2+ oscillations in pancreatic cancer PANC-1 cells. Biochem Biophys Res Commun 401: 154-158.

Reviewer 3 Report

I have several suggestions to improve the manuscript:

Localization of TLR2,4 and 9 play an important role in cytosolic calcium oscillations. Therefore clear visualization of TLR2,4 and 9 by immunofluorescence is important - Fig. 4C needs to be improved. Why there is no visible pattern in images with TLR4 or 9 alone in pancreatic acini and some pattern in merge images (Fig. 4Ca)? Fluorescence intensities of TLR2,4,9 (Fig. 4C) do not correspond to protein expression in Fig. 4B, e.g. according to WB image, TLR9 staining in acini should be the brightest.

No effect on basal calcium was found in the freshly isolated rat pancreatic acini (Fig 1). Could you demonstrate that TLR9 really colocalize with zymogen granule in pancreatic acini? It is not obvious from Fig. 4.

Extracellular histones are known contributors to cell damage. Is there some cytotoxic effect in used concentrations of Hx, Ach or CCK during the experiment?  

Is there some difference in viability or cell death induction of control and dexamethasone-treated AR4-2J cells before and after addition of H4?

Please provide in explicit form all available details in Material and Method section. E.g.: Western blot – better description, a composition of lysis buffer is missing; Immunocytochemistry – how the cells were fixed? Please, the description should correspond to a real order of individual steps (first blocation, then incubation with antibodies…)

Function significance of TLR9 down-regulation is clear, but there is not well documented this down-regulation by WB analysis. Fig. 7: WB image – no difference between TLR9i a NC in comparison with quantification.

Minor Essential Revisions

Explanations of abbreviations are missing in Abstract: e.g. CCK, Ach, TLR; in Introduction: M3.

Please don’t use 2nd antibody: Please add correct name of secondary antibody and number of a product.

Origin or preparation of mixed histones (Hx)? Missing description in Material and Methods.

2.7: fluorescence collected at 510 ± 25 nM…  - fluorescence is measured in nm.

Fig. 4: Expression of TLR2 mRNA in AR4-2J cells is lower than in acini but not according to quantification (Fig. 4Aa vs. 4Ab)

In Fig. 4Ba, the level of TRL2,4,9 is much lower in AR4-2J cells than is documented by quantification.

Author Response

I have several suggestions to improve the manuscript:

Localization of TLR2,4 and 9 play an important role in cytosolic calcium oscillations. Therefore clear visualization of TLR2,4 and 9 by immunofluorescence is important - Fig. 4C needs to be improved. Why there is no visible pattern in images with TLR4 or 9 alone in pancreatic acini and some pattern in merge images (Fig. 4Ca)? Fluorescence intensities of TLR2,4,9 (Fig. 4C) do not correspond to protein expression in Fig. 4B, e.g. according to WB image, TLR9 staining in acini should be the brightest.

TLR2 stained relatively well in pancreatic acini but not TLR4,9, but none of them were present on the peripheral plasma membranes. Our immunocytochemical experiments were done multiple times and sometimes in parallel, but only TLR2 stained relatively well, TLR4 and TLR9 less so in pancreatic acini. TLR2,4,9 all stained better in AR4-2J than TLR4,9 in pancreatic acini, which showed the peripheral plasma membrane localization of TLR9 very clearly in AR4-2J cells. Immunocytochemsitry is not normally used for protein quantification.

Both fluorescence merged images and further merge with bright field image were presented for better localization of the fluorescence structures.

We have now presented a larger version of Fig. 4 in the revised version of the MS.

No effect on basal calcium was found in the freshly isolated rat pancreatic acini (Fig 1). Could you demonstrate that TLR9 really colocalize with zymogen granule in pancreatic acini? It is not obvious from Fig. 4.

In an acinus the zymogen granule regions are roughly discernable under light microscope. TLR9 was found to be mainly localized in that region in the isolated rat pancreatic acini. In the now enlarged version of Fig. 4C in the revised MS one could see better. All original figure files are now up-loaded with the revised version of the MS.

Extracellular histones are known contributors to cell damage. Is there some cytotoxic effect in used concentrations of Hx, ACh or CCK during the experiment? 

Although in some experiments we applied CCK before histones in AR4-2J cells, in other experiments we used CCK after histones (Figs. 5, 7). After histones CCK still triggered calcium oscillations, which indicates that the AR4-2J cells were still healthy after histone treatments.

Oscillating (in intracellular calcium, membrane potential or other parameters) cells are more likely to be healthy. Check all the calcium traces in Fig. 3, even at the high histone concentration of 20 mg.L-1 calcium oscillations were observed in AR4-2J cells, the calcium concentrations returned to basal level after wash out of histones.

At higher concentrations of Hx in pancreatic acini, after Hx treatment, CCK or ACh failed to elicit calcium oscillations. As mentioned in Introduction, such histone concentrations were reported in either model animals or in patients. These histone concentrations exerted adverse effects. We are reporting in this paper that after histones, CCK or ACh no longer induced calcium oscillations. It is correct to say Hx is “toxic” because they quenched calcium oscillations; after histones CCK1 or M3 receptors no longer functioned as well as before histone treatment in pancreatic acini. We are reporting this inhibitory effect in the present paper. Cell surface receptors are likely the first targets for extracellular histone actions. 

Only physiological concentrations of CCK (picomolar) or ACh (nanomolar) were used in this work. For mixed histones, only short exposures (longest 30 min) were used, after histones exposure no simultaneous drops in F340 or F380 were observed (no loss of Fura-2).

Is there some difference in viability or cell death induction of control and dexamethasone-treated AR4-2J cells before and after addition of H4?

Only calcium oscillations rather than sustained calcium increases were seen, it is likely mainly physiological-like responses were observed. As for the differences before and after Dex treatment, Dex-treated cells were less likely to respond to histone, but all cells responded to CCK stimulation. We did not observe Fura-2 loss (a simultaneous drop in F340 and F380) with H4 treatment either before or after Dex.

Please provide in explicit form all available details in Material and Method section. E.g.: Western blot – better description, a composition of lysis buffer is missing; Immunocytochemistry – how the cells were fixed? Please, the description should correspond to a real order of individual steps (first blocation, then incubation with antibodies…)

We have now appended more details in Methods. Please refer to Sections 2.5 and 2.6. All changes are marked in red.

Function significance of TLR9 down-regulation is clear, but there is not well documented this down-regulation by WB analysis. Fig. 7: WB image – no difference between TLR9i a NC in comparison with quantification.

No complete obliteration of TLR9 was observed. Down-regulated TLR9 resulted in significantly reduced response to H4. The TLR9 down regulation (TLR9i) was significant when compared to control (CT), but no significant difference was seen between control (CT) and NC. This correlation has been supported by other experiments.

Minor Essential Revisions

Explanations of abbreviations are missing in Abstract: e.g. CCK, ACh, TLR; in Introduction: M3.

Abbreviations are now spelled out on first use both in Abstract and in the main text.  

Please don’t use 2nd antibody: Please add correct name of secondary antibody and number of a product.

Section 2.1 Para 1 Lines 13-15: the word “2nd” is now replaced with “secondary”, cat. no. for secondary antibodies are also provided.

Origin or preparation of mixed histones (Hx)? Missing description in Material and Methods.

Section 2.1 Para 1 Line 10: the abbreviation of Hx is now inserted: “Collagenase P, mixed histones (Hx) of calf thymus were from Roche (Mannheim, Germany).

2.7: fluorescence collected at 510 ± 25 nM…  - fluorescence is measured in nm.

Section 2.7, Para 1 Line 9: now corrected: 510 ± 25 nm.  

Fig. 4: Expression of TLR2 mRNA in AR4-2J cells is lower than in acini but not according to quantification (Fig. 4Aa vs. 4Ab)

No statistical difference was found, P > 0.05. Significant difference was found for TLR4, as indicated by the asterisk (*). We have now re-checked our data, Fig.4Ab is now replaced, but no marked changes in mean / SEM are noted. Only subtle / slight variations in the SEM bars are seen.

In Fig. 4Ba, the level of TRL2,4,9 is much lower in AR4-2J cells than is documented by quantification.

The error bars are relatively large, but no statistical significance was found.

Round 2

Reviewer 1 Report

The authors provided responses to most of my major concerns; however, the responses to Major point 1, 3, 4 and 8 did not adequately address the issues pointed out in my original review. Since Cells is relatively high impact MDPI journal, I believe these issues need to be addressed prior to publication to maintain the reputation of scientific rigor for this journal.

Concerns

Major point 1. - The authors suggest in their responses that the variation in the initial (basal) Fura ratios could be due differences in loading of Fura-2 among cells. Because Fura-2 is ratiometric Ca2+ indicator, and the authors are reporting a F340/380 ratio, differences in Fura are not responsible for the apparent differences in resting (basal) Ca2+ among cells.

Major point 3. - The authors suggest that differences in basal Ca2+ levels do not “appear” to affect stimulated responses; however, given the protein machinery involved in mediating Ca2+ influx and efflux and the functional dependency on cytosolic Ca2+ of these machinery, it is difficult to see how basal Ca2+ would not affect oscillation frequency and amplitude. At minimum, the authors need to reference published data suggesting that differences in basal Ca+ do not affect oscillation in their cell types.

Major point 4. – The RT-qPCR shown in the response document does not provide any replicate information and does not match the RT-PCR data shown in Figure 4 of the main text. The main text uses the RT-PCR to suggest that the pancreatic acini and AR4-2J cells express the TLRs at lower levels than pancreatic stellate cells. The qPCR figure provided in the response document does not show this information. Additionally, Figure 4 in the main-text suggests that TLR4 mRNA in AR4-2J cells is significantly higher than pancreatic acini; this is not recapitulated by the qPCR shown.

Major point 8 – The authors did not address this concern; fluorescence line scans are still the only evidence for TLR9 redistribution.

Author Response

The authors provided responses to most of my major concerns; however, the responses to Major point 1, 3, 4 and 8 did not adequately address the issues pointed out in my original review. Since Cells is relatively high impact MDPI journal, I believe these issues need to be addressed prior to publication to maintain the reputation of scientific rigor for this journal.

Concerns

Major point 1. - The authors suggest in their responses that the variation in the initial (basal) Fura ratios could be due differences in loading of Fura-2 among cells. Because Fura-2 is ratiometric Ca2+ indicator, and the authors are reporting a F340/380 ratio, differences in Fura are not responsible for the apparent differences in resting (basal) Ca2+ among cells.

 Indeed the Referee is correct that in an intracellular buffer solution containing pure Fura-2, the F340/ F380 ratios are dependent on the free calcium concentration only. But in a real cell, there are plenty of endogenous fluorophores which absorb also in the UVA (320 -400 nm) region and their absorption and emission spectra do not strictly conform to those of Fura-2 (Baier et al., 2006; 2007). Such UVA fluorophores may also vary in concentration in different cells. We agree with the Referee that resting calcium levels (about 100 nM) do not vary much in a healthy cell, but resting ratios of F340 / F380 do vary among cells and when measured with different batchs of cell.

There are other reasons that resting F340 / F380 levels may vary. We in the lab routinely use two calcium measurement systems as mentioned in the MS (Section 2.7 Para 1 Line -4). One of the systems used to show resting F340 / F380 levels up to 8 or higher (with a previous version of CCD, data obtained before the present MS, see [46, 49], but the other system always showed F340 / F380 ratios around 1. Since all experimenters in the lab rotate between these two systems, how does one compare among the experiments? At that time, we found it was necessary when doing data analysis and statistics to normalize all F340 / F380 ratios to the ratios in the initial 60 to 120 s in the same calcium traces (same cell normalization). See the following figure, for example [49].

Figure 2, from [49].

In the early days of calcium measurements, most labs converted their raw data from fluorescence or fluorescence ratios to absolute calcium concentrations. But people normally calibrate their system once a month. To do calibration each and every day mould pose quite considerable burden on the experimenter. Now people have accepted that the raw data might be a more accurate reflection of the real situations or changes in cell function than the converted and calculated concentrations.

Fluorescence measurements of cytosolic calcium may not be necessarily compared to electrical measurements of the resting membrane potential, for example, which might not vary much among the same type of healthy un-stimulated cells in millivolts. For calcium measurements, the raw data are a reflection of the absolute concentration. For people working with temporal changes of cellular calcium, the presentation of calcium concentration data as fluorescence ratios is acceptable. The resting ratios of F340 / F380 did vary among different cells and among different batches of cells, but by subtracting the basal levels in the integrated calcium peaks in the present work we found it was possible to do the statistical analysis among different experiments. Please note the relatively small standard error bars, in Fig. 3F, for example.

References

Baier J, Maisch T, Maier M, Engel E, Landthaler M, Baumler W (2006) Singlet oxygen generation by UVA light exposure of endogenous photosensitizers. Biophys J 91: 1452-1459.

Baier J, Maisch T, Maier M, Landthaler M, Baumler W (2007) Direct detection of singlet oxygen generated by UVA irradiation in human cells and skin. J Invest Dermatol 127: 1498-1506.

Major point 3. - The authors suggest that differences in basal Ca2+ levels do not “appear” to affect stimulated responses; however, given the protein machinery involved in mediating Ca2+ influx and efflux and the functional dependency on cytosolic Ca2+ of these machinery, it is difficult to see how basal Ca2+ would not affect oscillation frequency and amplitude. At minimum, the authors need to reference published data suggesting that differences in basal Ca+ do not affect oscillation in their cell types.

 Variations in resting calcium concentrations will affect cellular responses. Variations in F340 / F380 ratios might not as discussed in Major point 1 above.

We have as advised quoted 3 references and in Section 3.2 Para 2 Lines 7-9 the sentence now reads like this: “Note the varied resting F340 / F380 ratio levels among different cells, which have also been noted before, and are normalized when performing data analysis [46, 47, 49].” All modifications in the MS text are marked in red font.

Major point 4. – The RT-qPCR shown in the response document does not provide any replicate information and does not match the RT-PCR data shown in Figure 4 of the main text. The main text uses the RT-PCR to suggest that the pancreatic acini and AR4-2J cells express the TLRs at lower levels than pancreatic stellate cells. The qPCR figure provided in the response document does not show this information. Additionally, Figure 4 in the main-text suggests that TLR4 mRNA in AR4-2J cells is significantly higher than pancreatic acini; this is not recapitulated by the qPCR shown.

The pancreatic stellate cells were used as positive controls to make sure that our PCR primers etc were working because others have shown these cells expressed TLR2,4,9. These positive controls have already served their purpose.

In the main text Section 3.3 Para 1 Lines 5-7 we have now modified the previously inserted sentence to: “Real time quantitative PCR (RT-qPCR) was also performed, and it was found that TLR2,4,9 were all expressed but without significant differences in the expression levels between pancreatic acini and AR4-2J cells (data not shown).”

The purpose of these PCR experiments were to confirm that TLR2,4,9 were expressed at the mRNA level. Strict comparisons about their mRNA concentrations or their variations between pancreatic acini and AR4-2J were not attempted.

Major point 8 – The authors did not address this concern; fluorescence line scans are still the only evidence for TLR9 redistribution.

As we have stated in our previous response, cell fractionation experiments might not be easy to accomplish to reconfirm TLR9 transfer between plasma and intracellular membranes.

For real time monitoring of protein translocations often a fluorescent tag is appended to the N or C terminus of the target protein, in most cases such extension of the protein sequence is supposed to not affect the property or function of the studied protein, because the fluorescent tag is in most cases small (239 amino acid residues long for KillerRed, for example, see Jiang et al., 2017, and Ref 47). We were examining longer term (5 days) effects (by activating the nuclear glucocorticoid receptors) in the present work; conventional immunocytochemsitry would do and might be more relevant to the real physiological conditions fixed in time.

The TLR9 immunocytochemistry (ICC) data presented in Fig. 6 were clear and convincing, and the ICC data are consistent with data from calcium measurements. We were quite surprised initially to see such major changes in TLR9 location after dexamethasone (Dex) treatment. We could have added a GFP tag to the TLR9 then examine TLR9 translocation after Dex treatment, but we were observing here a rather chronic but not acute effect like calcium changes, such experiments may not add more to the MS than already are available.

We may in the future do some photodynamic modulation experiments with different versions of fusion protein KillerRed-TLR9 (for an example of tag fusion by genetically encoded protein photosentisisers, please see Ref 47, and relevant background information in Jiang et al., 2017) and check the effects of Dex treatment. The detailed investigations of such photodynamic modulation of TLR9 activity and their possible interactions with the CCK1 receptor or other calcium signaling plasma membrane proteins or intracellular transmembrane proteins in AR4-2J cells would be a separate project. We would like to thank the Referee for stimulating our interest in this area.

Reference

Jiang HN, Li Y, Cui ZJ (2017) Photodynamic physiology - photonanomanipulations in cellular physiology with protein photosensitisers. Front Physiol 8: 191.

Reviewer 3 Report

Thank you for better resolution and all changes concerning Fig. 4. However, I prefer to show the real pattern (arrangement) of TLR4 and 9 in pancreatic acini (Fig. 4C), which is described in the Results section, more than the explanation why these antibodies are less effective for immunocytochemistry. Could you please adjust the level of TLR4 and 9 to be visible localization pattern of these receptors which you are describing?

2.1 – Catalog No. of anti-TLR9 Ab is missing.
   - Catalog No. of Hx is missing.
I don’t understand the protein quantification in Fig. 4Bb. What is protein level 1, how it was calculated and how does it correspond with Fig. 4Ba. Please, add these information to Figure legend.
Line 256 : with P < 0.05 indicated with an asterisk (*) (Bb). – should be a description of Fig 4Ab, Legend of Fig. 4Bb is missing.

Author Response

Thank you for better resolution and all changes concerning Fig. 4. However, I prefer to show the real pattern (arrangement) of TLR4 and 9 in pancreatic acini (Fig. 4C), which is described in the Results section, more than the explanation why these antibodies are less effective for immunocytochemistry. Could you please adjust the level of TLR4 and 9 to be visible localization pattern of these receptors which you are describing?

Please download the original file of Figure 4. By slightly tilting and adjusting the LED monitor of your computer, you could see the fluorescent patterns more clearly. Please ask the editorial office for the original file of Fig. 4.

Alternatively, the source files for Fig. 4C TLR4, 9 are also re-adjusted as requested and presented below – kindly dim your room light for better effect.

TLR4 (above) and TLR9 (below) source files were processed by AimImageBrowser to change brightness from 50 to 55, contrast from 50 to 55; brightness from 50 to 65, contrast from 50 to 70, respectively.

2.1 – Catalog No. of anti-TLR9 Ab is missing.
   - Catalog No. of Hx is missing.

The cat. nos. are as follows: anti-TLR9 monoclonal antibody, ab134368; Hx, 10223565001. These cat. nos. are now added to the Materials Section 2.1. All alterations in the main MS text are marked in red.

I don’t understand the protein quantification in Fig. 4Bb. What is protein level 1, how it was calculated and how does it correspond with Fig. 4Ba. Please, add these information to Figure legend.

Software Image J was used to quantify the TLR and b-actin band intensity, and ratios of TLR / b-actin were then calculated and presented.

Line 256 : with P < 0.05 indicated with an asterisk (*) (Bb). – should be a description of Fig 4Ab, Legend of Fig. 4Bb is missing.

The legend for Fig. 4Bb now reads as follows: “TLR2,4,9 and b-actin bands in blots from ≥ 3 rats or ≥ 3 AR4-2J cultures were quantified by ImageJ, TLR2,4,9 protein levels were expressed as ratios of TLR / b-actin. No statistically significant differences were found (P > 0.05, Bb).”
